# A Review of Filtration Performance of Protective Masks

**DOI:** 10.3390/ijerph20032346

**Published:** 2023-01-28

**Authors:** Ao-Bing Wang, Xin Zhang, Li-Jun Gao, Tao Zhang, Hui-Juan Xu, Yan-Jun Bi

**Affiliations:** 1Hebei Key Laboratory of Man-machine Environmental Thermal Control Technology and Equipment, Filtration Performance and Environmental Health of Protective Materials, Xingtai 054000, China; 2Advanced Research Center of Thermal and New Energy Technologies, Hebei Vocational University of Technology and Engineering, Xingtai 054000, China; 3School of Chemistry and Chemical Engineering, Jiangsu University, Zhenjiang 212013, China

**Keywords:** personal protective equipment, masks, filtration efficiency, overall protective performance

## Abstract

Masks are essential and effective small protective devices used to protect the general public against infections such as COVID-19. However, available systematic reviews and summaries on the filtration performance of masks are lacking. Therefore, in order to investigate the filtration performance of masks, filtration mechanisms, mask characteristics, and the relationships between influencing factors and protective performance were first analyzed through mask evaluations. The summary of filtration mechanisms and mask characteristics provides readers with a clear and easy-to-understand theoretical cognition. Then, a detailed analysis of influencing factors and the relationships between the influencing factors and filtration performance is presented in. The influence of the aerosol size and type on filtration performance is nonlinear and nonconstant, and filtration efficiency decreases with an increase in the gas flow rate; moreover, fitness plays a decisive role in the protective effects of masks. It is recommended that the public should wear surgical masks to prevent COVID-19 infection in low-risk and non-densely populated areas. Future research should focus on fitness tests, and the formulation of standards should also be accelerated. This paper provides a systematic review that will be helpful for the design of masks and public health in the future.

## 1. Introduction

We are currently living in an environment where viruses and bacteria are everywhere; however, masks can protect people from being infected by viruses and bacteria as an effective means of physical prevention [1,2]. Some examples of viral outbreaks include the SARS virus, which broke out in 2003, the Middle East Respiratory Syndrome coronavirus, which was discovered in 2012, and the largest Ebola virus outbreak in the years 2014–2016. Severe Acute Respiratory Syndrome Coronavirus Type 2 (SARS-CoV-2) and its variants have continued to ravage the world since 2020 due to their high transmissibility and high global death rate [3,4,5]. Health management remains an extremely important topic worldwide. Masks as a basic means of protection have been shown to aid in protecting people’s lives.

The virus is transmitted through air-borne droplets or spray, which have been identified as the major routes for cross-infection with coronavirus 2019 (COVID-19) among human populations [5]. The function of a mask is to stop the spread of the virus, which attaches to aerosol airborne particles or respiratory fluid droplets. The evidence shows that it is useful for healthy people to wear masks in the community during a pandemic [6,7,8]. Wearing respirators at work is an effective way to avoid infection and death in healthcare workers [6,7,8,9], and hospital staff have been instructed to change their masks every two hours when conditions permit [9]. Airborne transmission of respiratory viruses is divided into exhalation, transport, and inhalation [10]. It recommended that HCWs wear medical masks when in contact with an infected person and cover their noses and mouths during coughing or sneezing with either a tissue or a flexed elbow [10]; otherwise, exhaled virions would continue to circulate in the air 10 s later without wearing a mask. Because the virus is deposited in aerosols and transmitted via speaking, coughing, or sneezing, a comprehensive understanding of the evolution of respiratory droplets is necessary to determine the transmission mechanisms of respiratory diseases [11]. Therefore, masks play a very important role in health management and future outbreaks of novel viruses [12].

Filtration efficiency is the most critical factor in the protective performance of masks, but the development of mask materials is the key to determining filtration efficiency. The materials used for masks are being continuously developed. Gauze masks were the earliest mask materials; early surgical masks mostly comprised 8 to 12 layers of cotton yarn [13]. The multi-layer dense accumulation of cotton yarn makes the fibers show a cross-pore structure, which can physically intercept larger particles or dust in the air. Cotton yarn also easily absorbs moisture, and the blocking efficiency is low [14]. In the 1960s, non-woven mask technology was born, and disposable masks were developed during the same period [15]. The masks are made using the melt-blowing technique, a process in which electrical charges are injected into the material, creating a quasi-permanent electric field that provides sufficient particulate matter (PM) filtration through an electrostatic force [16]. Non-woven fibrous substances such as polypropylene (PP), polyethylene (PE), polylactic acid (PLA), and polytetrafluoroethylene (PTFE) are the most studied materials [16,17,18]. Polypropylene melt-blown nonwoven fibers have stable properties, a fine fiber diameter (0.5–4 μm), high porosity, good air permeability, and good filtration resistance; therefore, the performance of the filter material prepared using polypropylene is much better than that of other materials [17,18]. Furthermore, studies have demonstrated that the electret filter (non-woven fibers) has a higher filtration efficiency, lower air resistance, and greater dust-holding capacity compared with traditional fiber filters [18,19]. Polypropylene nonwovens can also be electrized to give the fibers an electrical charge. The filtration efficiency of polypropylene non-woven fibers regarding dust, air bubbles, and viruses can reach 98.9%, and the filtration resistance is only 37.92 Pa under the effect of coulomb adsorption [18].

For masks, the research objectives have always included achieving a larger capacity, optimal comfort, efficient elimination of biological aerosols, and optimal filtering of air particles [20]. Those factors affect the final quality and performance of masks. Nanofibers (this term is explained in Appendix A, as are other terms) have large specific surface areas, small whole sizes, small weights, high permeability, and good pore connectivity; therefore, many researchers aim to improve the filtration performance of mask materials by preparing nanofiber structures [21,22,23]. In general, the filtering ability of a mask is affected by the specification of the mask filter and external factors [24,25]. In addition, studies have shown the importance of external factors such as surface velocity or airflow, stable or unstable patterns of flow, the state of charge of particles, respiratory rate, relative humidity and temperature, and loading time for mask filtration efficiency [26,27,28,29,30,31,32,33].

Nevertheless, the most important feature of virus protection is filtration efficiency; however, further research is needed. This is due to a lack of systematic reviews and summaries of the filtration mechanisms, classifications, influencing factors, and the relationships between factors and filtering performance. In this paper, the main contributions are as follows:

Firstly, the characteristics and filtering mechanisms of masks are reviewed.

Secondly, the main aspects of the research concern the factors influencing the efficiency of respirator filtration, including the aerosol particle size, type, gas flow, pressure differential, and fitness.

Thirdly, the regularity of the factors that influence filtration efficiency—aerosol particle size, type, gas flow, pressure differential, and fitness—are presented.

Finally, the filtration performance of masks is evaluated, the existing problems are summarized, and future development trends are proposed.

The related mechanisms of aerosol transmission, and the characteristics and classification of masks in this regard, are reviewed in Section 2, and the factors influencing the protective efficiency are reviewed in Section 3.1. The relationships between filtration efficiency and the aerosol particle size, type, gas flow, pressure differential, and fitness are discussed in Section 3.2. Section 3.3 evaluates the filter performance, price, wearing, and safety of masks. Finally, based on the existing problems in the development of masks, the future development trend is put forward in Section 4.

## 2. Mask Characteristics and Filtration Mechanisms

In order to study the factors affecting the filtration performance of masks and the recognition of relationships between factors and performance, it is necessary to understand the basic transmission mechanisms between aerosols and masks and the classification of masks to gain a clearer understanding. The mechanism of virus particles passing through fabric is divided into gravity sedimentation, inertial collision, interception, diffusion, and electrostatic interaction [34,35,36]. Section 2.1 discusses the types and characteristics of masks. The filtration mechanisms regarding the interaction between aerosols and masks are discussed in Section 2.2.

### 2.1. Types, Characteristics, and Differences among Masks

The mask pictures and fiber structure images are shown in Figure 1.

#### 2.1.1. Cloth Masks and Surgical Face Masks

(1)Cloth Masks

As one of the most basic types of masks, cloth masks are made from fibrous material, such as cotton, towels, pillow towels, and T-shirts, and can be homemade. A cloth mask and the cotton fiber structure can be seen in Figure 1. Cloth masks have relatively low filtration efficiency and limited protective effects and are generally recommended for protection against community transmission [8,37]. However, cloth masks cannot realize the efficient filtration of fine particulate matter, owing to the thick fiber diameters and large pore sizes. Improving the efficiency of these filters usually relies on increasing the number of mask layers, reducing the fiber diameter, and reducing and changing the fiber structure’s density, which lead to increases in respiratory resistance and discomfort from thermal and moisture perspectives, making them difficult to wear for a long time [38,39,40,41,42].

(2)Surgical masks

Surgical masks are not primarily designed to protect healthcare workers from airborne particles and are used to reduce bacterial spread from the mouth, nose, and face; however, the filtration efficiency of surgical masks is better than that of cloth masks [43,44,45]. The structure of a surgical mask can be described by SMS (Figure 1), consisting of spunbonded (the outermost layer), melt-blown (the middle layer), and spunbonded (the inner layer) layers, wherein the melt-blown layer is a filter layer (the middle layer), and the spunbonded layers comprise an outermost and an innermost waterproof layer [43,46,47]. Figure 1 shows the surgical mask and melt-blown polypropylene structure. The fabric structure of the melt-blown layers is denser than that of a cloth mask, and the fiber pores are smaller than those of a cloth fabric mask and other layers [17,37].

#### 2.1.2. N95, KN95 Masks, P100 Respirator Masks, and Powered Air-Purifying Respirators

(1)N95 /KN95 Masks

An N95 mask has a breathing valve through which viruses are removed from the air. According to the filter efficiency, oil resistance, and the characteristics of different national standards, KN95 masks have the same filtration efficiency as N95 masks for oil particles. An N95 mask consists of four different layers of filters. The innermost and outermost layers consist of nonwoven PP that is primarily hydrophobic to prevent water from being absorbed. The intermediate layer consists of modified acrylic supports to provide shape and thickness for the respirator, and the nonwoven melt-blown polypropylene layer (Figure 1) is used to trap unwanted particles [17,38].

(2)P100 Respirator Masks

P100 FFR masks are used to prevent the passage of toxic air particles in industrial environments where petroleum may be encountered [48,49]. The filtration efficiency of a P100 FFR mask is better than that of a N95 mask; nevertheless, the accumulation of water vapor and moisture affects the use of P100 FFR masks.

(3)Powered Air-Purifying Respirators

Powered Air-Purifying Respirators (AAPRs) are battery-powered blast devices that allow pressured air to pass through a filter (P100 or HEPA) into a full or half-face mask and can be used in place of an N95 mask [50,51,52].

### 2.2. Filtration Mechanism of Masks

Figure 2 shows the filtration mechanisms, mask classification, and SMS structure.

#### 2.2.1. Gravity Sedimentation and Inertial Impaction

The gravity sedimentation mechanism mainly works for large particles with a large density and with low airflow density. Larger particles are filtered out of the fabric via gravity as the air flows [53,54,55]. Maduna et al. mentioned that there is no obvious interaction between the fiber particles and the aerosol particles [55]. In this case, the particle size range of the aerosol particles is about 1 μm–10 μm [5]. Due to the effect of gravity on large aerosols, generally, for aerosols with particle sizes less than 0.5 microns, in order to facilitate the study of the filtration mechanism, the influence of gravity on the filtration mechanism can be ignored [56].

An inertial collision occurs when larger particles are captured by a fiber because of the fiber’s greater inertia. The reason for the collision is that the particle inertia is greater than the airflow resistance, and the particle deviates from the airflow line and is deposited on the surface of the fiber, resulting in a change in the direction of the particle movement in the airflow. With increases in particle size and wind speed, the amount of particle deposition increases. When the particle size is 1 μm or larger, the particle’s surface velocity and density are larger, and it is more easily captured by fiber particles; thus, inertial collision occurs [5,55,57].

For aerosols of nanoparticles or ultrafine nanoparticles, the inertial collision mechanism can be ignored. Wang et al. summarized the mechanism of nanoparticle filtration, mainly involving diffusion, interception, and electrostatic action under certain conditions [58]. Furthermore, they reviewed the mechanism of nanoparticle filtration and the effects of nanoparticle physical morphology, temperature, and humidity on the filtration performance, emphasizing the importance of studying these parameters [58]. In addition, there are some problems in the application of a basic filtration mechanism using nanoparticles, such as the thermal rebound of nanoparticles during filtration. Therefore, many scholars have studied the influence of thermal resilience on the filtration mechanism and efficiency of nanoparticles [59,60,61,62]. It is worth noting that the novel coronavirus particles are approximately 60–140 nm in size; however, the virus cannot exist independently and needs to be transmitted via attached droplets, which are about 5 μm in size [34].

#### 2.2.2. Diffusion and Interception

Brownian motion occurs easily when the particle size is 0.2 μm or smaller, and the particles perform random Brownian motions in the filter medium [63,64]. In this process, the particles do not move along the flow line but randomly collide with each other [65]. As a result, they are attracted by the fibers and deposited. As the particle size and facial velocity decrease, the diffusion rate increases, and the amount of deposition increases [66]. When the velocity is low, the residence time of particles through the filter medium is prolonged. Therefore, the probability of collisions between particles and the filter medium is greatly increased [54]. The particle diffusion mechanism with nanoparticles or ultrafine nanoparticles is more effective than the interception mechanism [67].

At first, the particles move with the airflow, but when the distance of the particles from the surface of the fiber is less than its radius, the particles interact with the fiber and are attracted, and the particles are directly intercepted by the fiber. This is what we call an interception mechanism. In general, interception mechanisms can successfully intercept aerosol particles below 0.6 μm [67]. It is noteworthy that the attraction for interception is the van der Waals force. In the interception mechanism, the smaller the particle size, the more obvious the interception; however, there is no obvious relationship between interception and a reduction in particle size [67]. When the size of the aerosol particles is between 100 nm and 1 μm, diffusion and interception mechanisms play a major role [53,68,69].

In the development of a filtration mechanism, inertia, interception, and diffusion are important mechanisms for removing fiber aerosols [70]. Shi et al. analyzed the single and composite mechanisms involved in isolated rotating fiber filtration, which mainly include inertia impaction, interception, and diffusion [70]. Lee et al. conducted a theoretical analysis of the filtration mechanism and derived the expressions of the diffusion and interception of the neighboring fibers in the region with the highest filtration efficiency, but the applicability of this work needs to be verified [71]. Studies have shown that in the maximum particle size penetration region, diffusion and interception mechanisms play a major role; therefore, only diffusion and interception mechanisms are considered in Lee’s paper [71].

Compared with the existing studies [5,72], we summarize the applicable sizes and main driving forces of aerosol filtration mechanisms in Table 1, which conveniently demonstrates how the driving force differs according to the sizes of the aerosols in different filtration mechanisms.

#### 2.2.3. Electrostatic Attraction

This capture mechanism is divided into the electrostatic gravity mechanism and the mechanical filtration mechanism (gravity, interception, inertial collision, and the diffusion mechanism mentioned above) [73,74,75]. When aerosol particles and fiber particles have opposite charges, they attract each other due to electrostatic attraction, and this is a very effective filtration mechanism in fibers, especially for very small particles [5,34]. Electrostatic attraction can capture large and small particles from the airflow. As for nanoparticles, they can easily slip through the gaps between the fibers. Electrostatic attraction mainly serves to remove low-mass particles that are attracted or bonded to the fibers [76]. Gravitational subsidence, inertial impact, diffusion, and interception mechanisms do not take into account the effect of electrostatic attraction on particles. One technique used to enhance the effects of these mechanisms is electrostatic force, which is especially useful for particles of moderate size [76]. For sub-micron particles (0.1–0.5 μm), the electrostatic filtration mechanism is more important than the mechanical filtration mechanism [77].

If electrostatic attraction is increased, a filter’s efficiency is improved, so under the condition of a lower packing fraction, the filter’s efficiency can reach a certain value, thus reducing air resistance. Electret filters are composed of permanently charged fibers, which makes electret filters suitable for high efficiency and low resistance in air-cleaning applications [77]. Electrets consist of dielectric materials that, when subjected to an electric field, produce a quasi-permanent charge. There are three methods of generating electret air filters: (i) corona charging, (ii) friction charging, and (iii) induction charging [78,79,80].

Although electrets can improve filtration efficiency, their charge decays with time. Lee et al. found that electrets’ exposure to high temperature, humidity, and solvents leads to a decline in their filtration performance [81]. Water molecules absorbed in electrets contribute to electricity conduction, and similarly to heat treatment, water molecules may affect the mobility of dipoles and carriers [81]. In order to enhance the stability of electrets’ charge, phase change materials with thermal stability and water resistance were developed [82,83,84].

## 3. Influencing Factors and Protective Performance of Masks

The present research focuses on improving the filtration efficiency of masks [53,85,86,87,88,89,90,91,92,93,94,95,96,97,98,99,100,101,102,103]. Masks’ protective performance largely depends on two significant factors: filtration efficiency and fit (facepiece leakage). However, filtration efficiency is related to many factors. The factors affecting filtration performance and testing methods for masks are summarized in Section 3.1. The relationships between the influential factors—the aerosol particle size, type, gas flow, pressure differential, and fitness—and filtration efficiency are presented in Section 3.2. The filtration performance of masks is evaluated in Section 3.3. The factors affecting the filtration efficiency of masks include the flow rate, surface velocity, aerosol particle type, particle size, temperature, filtration parameters (thickness, fiber diameter, and fiber packing density), decontamination filtration efficiency, and fit. The detection of mask filtration performance is of great significance for protecting individuals from viruses and bacteria. Although there are many experimental methods for measuring the filtration efficiency of masks, there is no uniformly applied method in the current experimental research, owing to diverse research assumptions and objectives [85]. The current experimental methods and the factors affecting the filtration performance of masks are summarized in Table 2 [53,86,87,88,89,90,91,92,93,94,95,96,97,98,99,100,101,102,103].

### 3.1. Factors Affecting Filtration Performance of Masks

#### 3.1.1. Aerosol Particle Size and Type Factors

Due to the dependence of filtration efficiency on the size of particles, in general, the sizes of the aerosols used in experiments differ according to research purposes. For the different types of masks, many types of aerosol particles are reported in the literature, including bioaerosols (usually bacterial) and non-biological aerosols (environmental particles, aerosols from diesel combustion, or others, such as salt). In the experimental studies on filtration efficiency, NaCl aerosols are most commonly used, and the sizes of the aerosols vary from 10 to 1000 nm (Table 2). Regarding bioaerosols, the MS2 virus (10−80 nm) was used as a nonharmful simulant for several pathogens in an experimental study [86,92].

The test for N-series masks with higher filtration efficiency (NIOSH Title 42 Specification, Part 84 of Federal Regulation) specifies an aerosol particle size of approximately 300 nm, which is the most penetrable particle size (MPPS) [104]. N95 masks have a filtration efficiency of at least 95% for NaCl particles 100–300 nm in size. When the certified flow rate is 85 L/min, the filtration efficiency reaches 99.5% or higher at about 750 nm. The removal rate for environmental particles with submicron particles smaller than 750 nm can be up to 100%. The maximum penetration mass fraction of particles smaller than 750 nm was found to be 1.8% [67]; however, the respirator penetration rate was less than 5% at 30 L/min, while at 85 L/min, it was more than 5% [105]. For virus samples, N95 masks did not provide the expected level of protection against small particles in the 10–80 nm range, especially at higher inhalation flow rates. The filtration efficiency of N95 masks averaged at 94.4% when the particles were approximately 50 nm in diameter [106,107]. In another study, the mask material was tested at 0.3 mm, and the particle filtration effect was more than 95% [108]. It was certified that N95 masks could protect wearers from particles 300 nm and above in size. Aerodynamic sizes of 800 nm or greater have been reported for airborne bacteria [109].

For surgical masks, the ASTM PFE test uses an unnaturalized latex pellet aerosol with a particle size of approximately 0.1 μm, a flow rate of 28.3 L/min, and relative humidity (RH) at 30–50%, as specified in the standard, and the filtration efficiency of surgical masks should not be less than 95% under these conditions [110]. Beyond that, the NaCl aerosol is the best alternative for distinct experimental operations whether the mask type is N95, a surgical mask, or another personal protective device.

#### 3.1.2. Airflow Rates and Pressure Differential Factors

The air velocity should be set to simulate the breathing intensity of the human body in actual environments, ranging from that under high workloads (300−400 L/min) to low intensities (42.5 L/min and 85 L/min), and with average exhaustive ventilation rates as high as 114 L/min [111,112,113,114]. The constant flow rate for the test is 85 L/min or 42.5 L/min as specified in the test standard of NIOSH, where the 85 L/min ventilation rate is for a single-filter respirator and 42.5 L/min is for a double-filter respirator. However, the high ventilation rates and high peak flow rate can only be maintained for a short time under the ultimate working load, and the value is higher than the constant 85 L/min gas flow rate specified by NISOH [115].

The pressure differential is the pressure difference between the two sides of the material or fabric under a certain air velocity, and the pressure difference, or drop, reflects how easy it is to breathe through the filtered material. A low pressure differential indicates that the gas passes through the material more easily and that breathing is easier. The surface velocity has a certain relationship with the pressure on both sides of the fabric, whereby reducing the air velocity will reduce the pressure difference and increasing the thickness of the filter material will increase the pressure differential. In the NIOSH test, the filters must achieve a minimum of 95% FE at a differential pressure no higher than 1.15 w.c. for an N95 mask [116]. ASTM International sets standards at 20% FE at no more than 0.59 w.c. airflow resistance for non-medical fabric masks [117], and ASTM International states that the airflow resistance should ensure at least a 95% FE for a surgical mask and a differential pressure of no more than 0.90 w.c. under similar testing conditions [118].

#### 3.1.3. Mask Fitness Factors

The fit of a mask has an impact on its filtration efficiency, so the fit is tested to determine whether the mask fits the wearer’s face or leaks particles inward (the total leakage of particles through end seals, valves, and gaskets, and penetration through filters). In a quantitative fitness test, the general approach is to measure the concentration of particles inside and outside the mask as the wearer performs a series of exercises. Typically, fitness tests can quantify the outward leakage of aerosol particles, described by the fit coefficient, which is the ratio of the particle concentration outside the mask to the particle concentration inside the mask. Unfortunately, there are only procedures for filtration efficiency certification, and leakage-related certification has not been proposed [119]. N95 masks, surgical masks, and cloth masks are not tightly designed. Generally, N95 masks must pass a fitness test before being worn by medical workers. A mask with a fitness factor greater than 100 can pass the fitness test. However, at present, 100% of surgical masks and cloth masks fail the fitness test [17].

In addition to basic performance, the discomfort caused by the accumulation of heat, humidity, and carbon dioxide during actual wear are also factors that affect the filtration efficiency of a mask. Increasing electrostatic attraction and decontamination methods are also factors affecting the filtration efficiency of masks. A detailed discussion of the factors affecting filtration efficiency and regularity is presented in Section 3.2.

### 3.2. Protective Performance and Discussions

#### 3.2.1. Effects of Aerosol Particle Size and Type Factors

From the development of ordinary gauze masks to the current respiratory masks, filtration efficiency has been improved. Many researchers have evaluated the protective performance of masks including filtration efficiency with different aerosol sizes and types and face velocities (air velocities). The filtration efficiency of masks is an important performance index for measuring whether a mask can be qualified or not, and the improvement of filtration efficiency has always been the goal of mask researchers. Among the vast literature on filtration efficiency, researchers’ contributions on the filtration efficiency of masks mainly include comparative studies on cotton masks, homemade masks, medical masks, and N95 masks. It should be noted that a permeability index is also used to measure the filtration efficiency of a mask, and the sum of the permeability and filtration efficiency is 1. We reviewed some studies on the filtration efficiency of common fabric materials, medical masks, and N95 masks; examples of the relevant studies are provided here [27,30,53,86,87,90,92,93,101,120,121,122,123]. Different researchers chose different aerosol sizes and types when studying the filtration efficiency of different types of masks. On the basis of these, the regularity of filtration efficiency with regard to the aerosol type and size and surface velocity was obtained. We studied ordinary fabric masks, medical masks, surgical masks, and N95 respirators with regard to these three factors.

The relationship between particle size and filtration efficiency is not linear, and filtration efficiency generally decreases with a decrease in aerosol size. However, some studies show that filtration efficiency may increase or decrease with an increase in aerosol size, and this law is not constant [53]. Common fabric materials can also play a role in preventing the spread of disease. However, common fabric materials can only provide marginal protection; the data showed that the FEs for masks based on fabric materials, such as homemade masks and T-shirts, are in the range of 3–33% for NaCl aerosol particles 1000 nm in size and a face velocity of 5.5 cm/s [101]. A cotton mask’s FE is 99.5% and 98.8% for bacteriophage MS2 at sizes of 6.0 µm and 2.6 µm, respectively. A dried disposable pocket baby wipe’s FEs are 98.5% (6.0 µm) and 97.6% (2.6 µm). A three-layer hemp poly membrane and cheesecloth’s efficiencies are 93.6% (6.0 µm) and 80.9% (2.6 µm), and a Victorian DHHS mask’s are 98.6% (6.0 µm) and 99.1% (2.6 µm) [92]. Contrary to this conclusion, the literature indicates that the filtration efficiency of masks decreases when the particle size range is larger, according to tests using wide-ranging particle sizes [93], which showed that the filtration efficiency of 6 types of fabric decreased with an increase in aerosol size (25–200 nm). Homemade masks’ EFs are 84.54% (6–200 nm) and 90% (6–89 nm).

The effect of the aerosol type on filtration efficiency depends on the type of aerosol used in the experiment, which represents the specific type of virus or bacterium. The bacterial species described for aerosol experiments [86] are related to reference experiments, which show that the EF for B atrophaeus is higher than that for MS2 for the same type of mask (Figure 3a). The research showed that the filtration efficiency decreased with a decrease in aerosol particle size (Figure 3b) [92]. A surgical mask’s FE is 53–75% for NaCl aerosol particles < 300 nm [94], which is not in line with the standard regulations [124], and its FE is 95% for KCl aerosol particles (with solution concentrations of 0.02–0.0025) [98] and 98.1% for NaCl aerosol particles (0.02–3.00 μm) [99], which is also not in line with the standard regulations, at least in terms of filtration efficiency. A surgical mask’s FE is 70–83% when the particle size is 0.1μm, and its FE is 74–92% with 1.3 µm polystyrene latex (PSL) [88]. Furthermore, increasing the number of layers of fabric in a mask can also improve the filtration efficiency. When medical masks are made by adding paper to the inner surface, the increase in EF is 7% [93]. However, the filtration efficiency exhibited a marked decline when the layer was inserted into a two-layer cotton mask (Figure 3d) compared with the measured base filtration efficiency (Figure 3c), and this may be explained by fitness issues, as explained in detail in Section 3.2. There is no doubt that N95 respirators have a higher filtration efficiency than ordinary fabric respirators. The EFs of N95 respirators and surgical masks decrease with an increase in particle size, as shown in Figure 3b. The N95 masks’ FE decreases as the surface velocity increases (Table 3).

In addition, homemade masks are indispensable in the prevention of community transmission, partially relieving the pressures on the community, but they should be used as a last resort in preventing droplet transmission from infected individuals because the EF of surgical masks is three times as high as that of homemade masks [86,92,93]. Current commercial non-medical masks do not meet the new standards, so maintaining a healthy and safe distance, apart from wearing a mask, is necessary [90]. Studies tend to conduct comparative tests of mask filtration efficiency, and some of them make different types of masks to facilitate the tests. They show that the filtration efficiency of homemade cotton masks is undoubtedly much lower than that of N95 masks, medical masks, and disposable masks. From the literature, we found that the filtration efficiency of cloth masks and homemade masks was very low (10%–30%) [86,101], while that for medical masks was at least 90%, and that for N95 masks was at least 90% [86].

#### 3.2.2. Effects of Airflow Rates and Pressure Differential Factors

The filtration efficiency of ordinary fabrics used in most experiments decreases with an increase in surface velocity, except for some materials (Table 3). In the case of N95 masks, the filtration efficiency values are 99.88% and 96.60% when the face speed is 5.5 cm/s and 16.5 cm/s, showing a significant decrease [101]. The filtration efficiency of four common cloth masks (Respro Bandit, Breathe Health Cloth, and Today’s gentleman) increased with an increase in face speed, and the Breathe Health Fleece filtration efficiency stayed the same when the face velocity ranged from 5.5 cm/s to 16.5 cm/s. Similar results were found in the research by Karin [91], where the filtration efficiency at a low flow rate was higher than that at a high flow rate. The NANO KN95 filtration efficiency was 100.0% at a low flow rate and 99.7% at a high flow rate.

The pressure difference increases and filtration efficiency decreases at a high flow rate. Thus, an increase in filtration efficiency means an increase in respiratory resistance. NANO KN95 masks have a differential pressure of 77 pa at a high flow rate and 36 pa at a low flow rate, which means that the filtration efficiency value is reduced by 3% [91]. Liu et al. studied the relationship between filtration efficiency and respiratory resistance. The filtration efficiency of mask B was the highest among the three types of masks, but its respiratory resistance was the lowest [89]. Mask B (a medical mask) has both a high filtration efficiency (70%) and low respiratory resistance (90 pa) and has the lowest porosity among the three masks. In addition, the pressure difference of N95 masks is greater than that of medical masks [53]. In general, an increase in filtration efficiency will inevitably lead to an increase in respiratory resistance, which is the main factor restricting the improvement of the filtration efficiency of masks.

Generally speaking, filtration efficiency decreases with an increase in the gas flow rate, and the pressure difference correspondingly increases at this time. In addition, an improvement in filtration efficiency means an increase in respiratory resistance, which is a universally recognized problem.

#### 3.2.3. Effects of Fitness Factors

Fitness plays a key role as one of the factors affecting the filtration efficiency of masks, and poor tightness inevitably reduces the filtration efficiency of masks. The main reason for decreases in filtration efficiency is related to the fit degree, and quantitative fit tests have been unable to distinguish between particles permeating the filter media and particles leaking through poorly fitted areas [87]. The permeability of the filter material plays a role in places where particles are likely to move through loose masks, especially where smaller particles may easily follow the air vector around an imperfect fit [125,126]. Thus, we discuss the recent research and emphasize the significance of fitness for masks [53,87,88,91,92,94,95,97,100]. Figure 4a shows quantitative fit testing results for the new N95 respirator after oven and steam heat-treatment cycles [103]. The tester went through several motions, as follows: “Normal Breathing, Deep Breathing, Head Side to Side, Head Up and Down, Talking and Bending Over Normal Breathing”.

Fit issues and leakage can lead to a 60% decrease in EF while allowing the exhaled air to vent efficiently [53]. When inserted into a two-layer cotton mask, all the filter materials exhibited a significant drop in filtration efficiency compared with the measured base filtration efficiency. In this test, 3M 8511 and KN95 respirators as well as medical and dust masks were tested in mask form, whereas the remainder were tested by inserting them between two layers of cotton masks. The filtration efficiency of a KN95 mask sealed to the head before testing with a thermoplastic adhesive was very close to its basic filtration efficiency, proving that the main reason for the decrease was the quality of wear [87]. The factors of fit and face seal leakage were further highlighted in a study on N95 masks [88], and future research is needed to explore the optimum design for ensuring proper fit [92].

The N95, KN95, and HEPA filters passed the fit tests; however, the homemade mask, Montana mask, Halyard H600, and WypAll X80 failed the fit tests. Good fit is undeniably as important as high performance, and most of the studies on masks demonstrated stable filtration efficiencies over 2 h [91]. In total, nine types of surgical masks with a normal breathing fit failed [97]. Quantitative intercomparable data for the designs/fits of masks and materials are provided in [94]. The face seal leakage-to-filter ratio of the surgical mask ranged from 4.8 to 5.8, and that of the N95 mask was 7-fold (0.04 µm), 10-fold (0.1 µm), and 20-fold (1 µm) higher. The face seal leakage ratio was mainly an issue for the tested respirator/mask, and facial/body movement had a pronounced effect on the relative contribution of the two penetration pathways [95]. The leakage was 26% (10 L/min) and 11% (70 L/min) for N95-A2 masks, SMs and FPUs (electret layers), which is less than that for N95 masks [100]. Future studies will focus on experimentally examining gaps and leakage with different manikins donning the forementioned brands of PPE [100].

### 3.3. Evaluation

The protective properties of masks vary according to the type of mask. The filtration efficiency, durability, occasion applicability, reusability, respiratory resistance, prices, and leakage-proof abilities of different types of masks are summarized in Figure 5.

The respiratory resistance of cloth masks, surgical masks, N95 masks, P100 FFRs, and PARRs increased as the filtration efficiency increased. Many scholars have created evidence-backed cloth masks to reduce the transmission of viruses such as COVID-19 [86,127,128,129,130,131]. Surgical masks are three times more effective at blocking transmission than homemade masks, and cloth masks are more effective than no mask at blocking the spread of the virus when masks are in short supply [86]. A researcher found that the filtration efficiencies of cloth masks and medical masks were 80%; the cloth masks were made of 100% polyester microfiber [132]. Sterr C.M. et al. found that cloth masks were the least effective at filtration, filtering out less than 20% of the tested aerosols [7]. These studies fully demonstrated the usefulness of cloth masks in preventing the spread of the virus. Respirator masks are personal protective devices used to combat high risks of respiratory infection and can filter small particles and prevent leakage around mask edges during inhalation, making them distinct from surgical masks, and it is more appropriate for confirmed patients to wear respirators [133,134]. N95 masks are respirators with 95% filtration efficiency and can strongly prevent the transmission of respiratory diseases [108]. The increased tightness of an N95 mask and the reduced porosity of the material fabric of the filter layer make the filtration efficiency better than that of surgical masks, but the respiratory resistance is higher than that of surgical masks. The filtration efficiency of P100 FFRs and AAPRs is at least 99%, and the respiratory resistance is higher than that of N95 masks [50,51,52,135].

Furthermore, different types of masks are worn in different contexts. Cloth masks are used when the risk of infection is relatively low, such as in non-densely populated communities and outdoor areas. In addition, the public is advised to wear cloth masks amid a shortage of masks [86,127,128,129,130,131]. It is very important for the public to wear surgical masks advocated by researchers for the prevention of COVID-19 when not in close contact with influenza patients or suspected influenza patients [136]. Surgical masks are used in areas of relatively high risk, such as streets, supermarkets, and offices with fewer staff. Healthcare workers who wear surgical masks are at double the risk of infection compared with those who wear respirators [137]. Consequently, if there are high-risk areas such as operating rooms, wards, buses, and train stations with confirmed cases, N95 masks or other respirator masks (P100 FFRs and AAPRs) are essential to reduce the risk of infection [137]. A PARR is suitable for sites with high concentrations of infectious aerosols, especially when the risk is unknown or uncertain.

In addition, issues with leakage, efficiency, and thermal comfort arise, which affect the suitability of masks. The tightness of cloth masks is very poor, and they fail fitness tests. N95, P100 FFR, and AAPR masks have the advantages of good face sealing, no face leakage, and reduced filter penetration of aerosol particles, as well as passing fit tests. Furthermore, the N95 masks P100 FFR and AAPR can be reused after decontaminating them using approved techniques [138]. On the one hand, wearing N95 and P100 FFR masks for a long time produces a lot of water vapor, and water accumulates, causing a decrease in filtration efficiency; on the other hand, the use of N95, P100 FFR, and AAPR masks for different people need to be researched. Moreover, AAPR masks can not only be reused and cleaned but also have good thermal comfort and can be worn appropriately for a long time.

These masks (cloth, N95, P100 FFR, and AAPR masks) are available for purchase on the market. PARRs are expensive and have many parts; thus, it takes time to assemble and check the filters frequently. The batteries need to be charged frequently, and the noise generated by the airflow during use can be uncomfortable [50]. N95-type air-purifying respirators are convenient, easy to use, and low-cost [134]. The price of a PARR depends on its accessories, and a respirator with basic parts and protective effects costs about USD 40. The market prices for N95 masks and surgical masks are around USD 2.2, USD 0.076, and USD 6.7. If one chooses to wear an N95 mask or a surgical mask, one can expect to pay between USD 803 and USD 186 in a year.

In summary, wearing a mask can keep us safe from COVID-19 when in contact with COVID-19 carriers in a dangerous environment (Figure 6). The particle filtration efficiency of N95 masks was more than 95% with NaCl particle sizes of 0.3 mm. However, for real 10–80 nm viruses, the filtration efficiency is not ideal at high suction flow rates, and the filtration efficiency is less than 95% (for 50 nm viruses) [106,107]. The size of particles absorbed by the virus varies according to the mode of transmission. COVID-19 virus airborne aerosol particles are smaller than 5 μm, and respiratory fluid droplets are between 10 and 5 µm [139], and aerodynamic sizes of 800 nm or greater have been reported for airborne bacteria [109]. Because P100 FFR and PARR masks are more efficient at filtering, N95, P100 FFR, and PARR masks could, therefore, protect wearers from particles of 300 nm and above, which is enough to filter out COVID-19. The filtration efficiency of surgical masks should not be less than 95% when the unnaturalized latex pellet aerosol particle size is 100 nm and the gas flow rate is 28.3 L/min [110]. Consequently, surgical masks and N95, P100 FFR, and PARR masks can protect people from COVID-19.

## 4. Problems and Developments

### 4.1. Problems

Based on the analysis of the factors that affect filtration efficiency discussed above, the existing problems that affect performance are summarized. First of all, the problem of tightness is the most prominent. Especially in actual use, there is no relevant standard; therefore, relevant standards need to be formulated. Secondly, the design of masks should consider the issue of tightness. It is suggested to collect information on the face shapes in different regions of the world with the help of new technological methods so that the shape of the face can be considered in the design of masks, making them suitable for varied face shapes, in order to increase the tightness of fit. Finally, in the experimental studies on the measurement of mask filtration efficiency in the literature, the use of a wide range of masks is reported, but there are no uniform standards or requirements for mask materials, so it is difficult to compare the results of different studies in detail. Therefore, it is proposed to use a uniform mask material for testing. For new research and development materials, the processes by which the materials are prepared should be specified in the published articles in order to help future researchers to repeat the process.

### 4.2. Future Developments

Improving the protective performance of masks and reducing pollution in the environment will be future development directions for masks. Fabrics or masks with high filtration efficiency and outstanding air permeability are constant pursuits. Furthermore, the problem of tightness should be addressed and masks fitting different face shapes should be developed. In addition, ensuring thermal comfort when wearing masks will also be a direction for future developments for human health. Effectively transferring the water vapor inside a mask while maintaining the filtration efficiency is a technical problem to be solved. In addition, in today’s era of serious environmental pollution, biomass mask materials are a development trend, which could not only meet the requirements of environmental protection, but also reduce the environmental load. Among the many research and development directions for mask materials, solving the problem of air permeability and improving the protective effect are the most important issues.

## 5. Conclusions

In the context of COVID-19, there is a paucity of systematic summaries on the filtration performance of protective masks. In this paper, the influencing factors and protective performance of masks are summarized. The conclusions are as follows:

Aerosol filtration mechanisms are divided into gravity sedimentation, inertial collision, interception, diffusion, and electrostatic interaction. The results for multiple mechanisms and a mechanism’s type of interaction are related to the aerosol particle size, airflow density, and material structure. However, in actual filtration mechanism processes, inertial collision, interception, and diffusion are important mechanisms for removing fiber aerosols, which is complex. Improving the efficiency of a filter is dependent on increasing the number of mask layers, reducing the fiber’s diameter, and reducing and changing the fiber’s structural density.

The influence of the aerosol size and type on filtration performance is not constant. Filtration efficiencies may increase with an increase in aerosol particles or decrease with a decrease in aerosol particles. However, generally speaking, for the same type of mask, the protection performance should improve with an increase in aerosol particles. The type of aerosol used is specific for a specific environment, and the aerosol used in a laboratory is engineered to not be harmful to people. The effect of gas flow rates on filtration efficiencies is very clear. The filtration efficiency decreases with an increase in the gas flow rate. The higher the filtration efficiency, the greater the pressure difference and increase in respiratory resistance. In addition, matching is a common problem observed in masks, and masks have improved matching performance with higher filtration efficiencies.

The respiratory resistance of a cloth mask, surgical mask, N95 mask, P100 FFR mask, and PARR mask increased in the listed order and as the filtration efficiency increased. In general, an increase in filtration efficiency will also lead to reduced comfort. However, if the thermal comfort performance of masks is improved, the cost and price of masks will increase. Therefore, on the basis of balancing the protective function and economy, it is recommended that the public should wear surgical masks to prevent COVID-19 infection in low-risk and non-densely populated areas.

Most researchers have focused on filtration efficiency and ignored the overall protective efficiency of masks. The filtration efficiency alone cannot explain the facial seal leakage and the fitting degree of a mask for a wearer. Improvements are urgently needed for the degree-of-fit standards in order to improve the overall protective performance. In the future, it will be important to improve the matching degree and filtration efficiency of masks. Moreover, respiratory resistance should be controlled relative to the minimum acceptable range of the human body. In addition, biodegradable biomass mask materials also represent a future development trend because the full utilization of biomass energy could solve the environmental problems caused by the mass use of masks.

In conclusion, this paper provides detailed data to support experimental mask research, ideas for mask designers, a theoretical basis for the improvement of mask protective performance, and assistance for the formulation of standards. In addition, it contributes to human public health.

## Figures and Tables

**Figure 1 ijerph-20-02346-f001:**
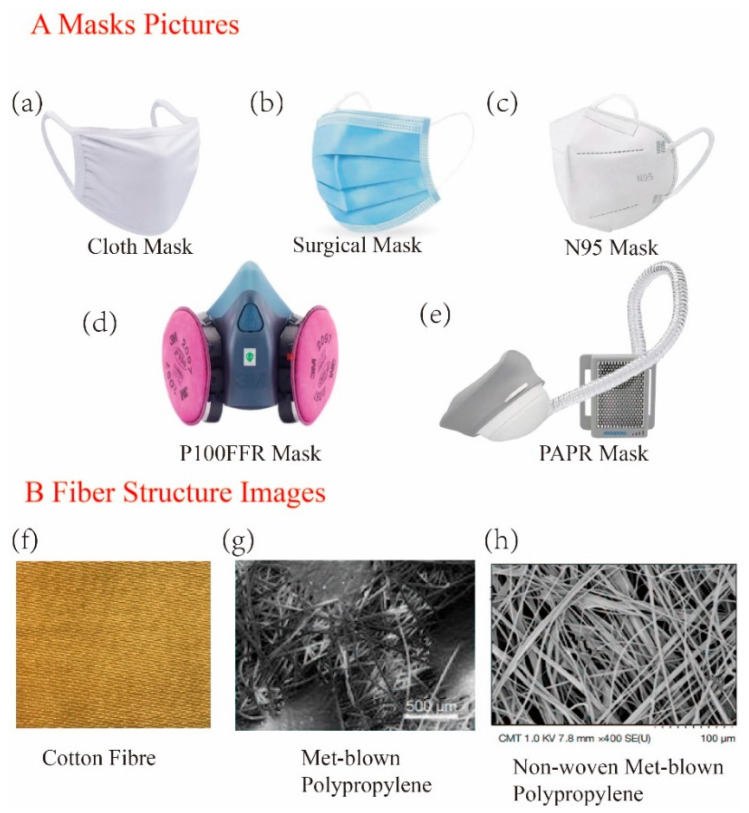
Mask pictures and fiber structure images [17,37,38]: (**A**) represents the mask pictures, including cloth masks, surgical masks, N95 masks, P100FFR masks, and PAPR masks; (**B**) shows the fiber structure images and cotton fiber, melt-blown polypropylene, and non-woven melt-blown polypropylene materials that are used in masks.

**Figure 2 ijerph-20-02346-f002:**
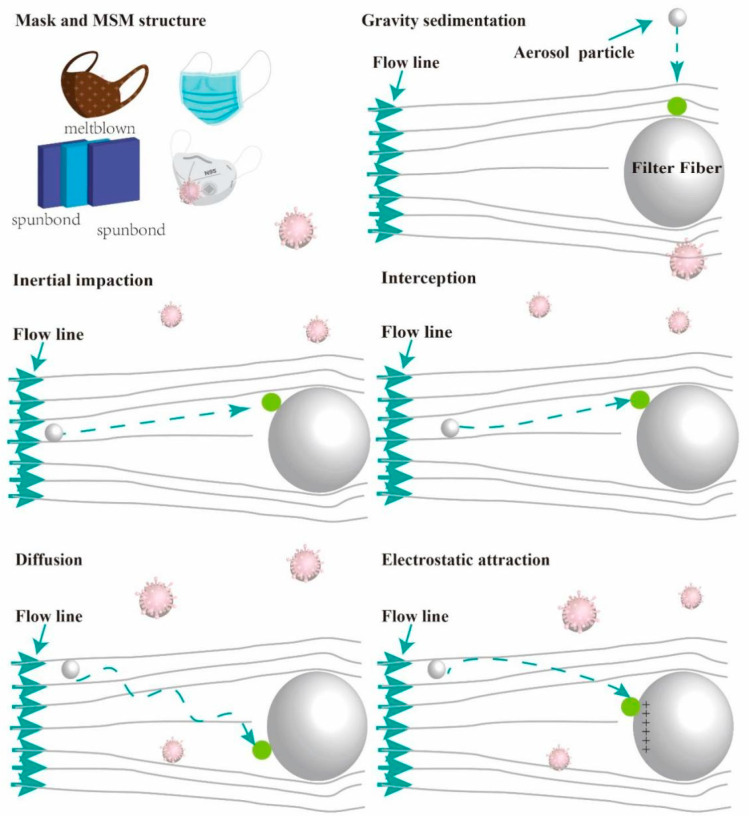
Filtration mechanisms, mask classification, and SMS structure.

**Figure 3 ijerph-20-02346-f003:**
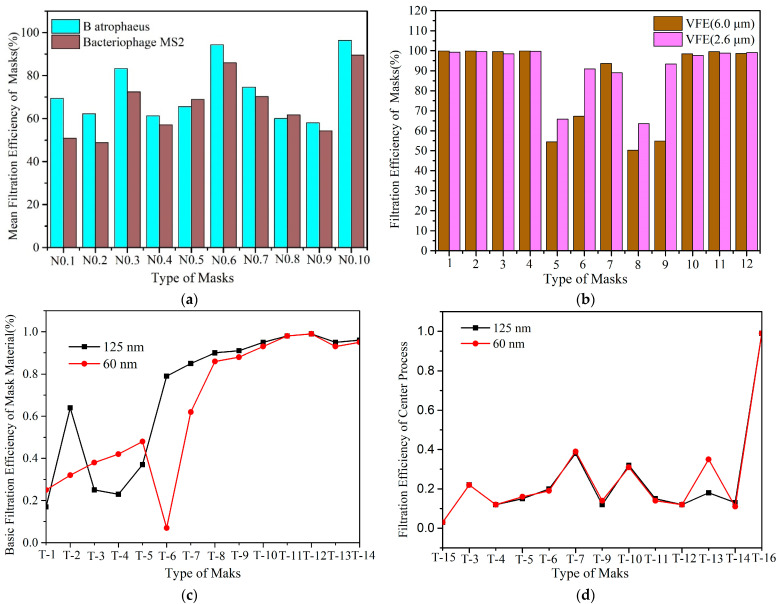
Effects of aerosol particle size and type on filtration efficiency: (**a**) Comparison of filtration efficiency between surgical masks and 9 types of fabric mask. N0.1–N0.10 represent 100% cotton, scarf, tea towel, pillowcase, antimicrobial pillowcase, vacuum cleaner bag, cotton mix, linen, silk, and surgical mask; these categories of masks are derived from those defined in the literature [86]. (**b**) The filtration efficiencies of 6 different types of fabrics were compared for 2 aerosol sizes, 6.0 μm and 2.6 μm, respectively. Numbers 1–12 represent the N95, surgical 1, surgical 2, disposable 1, fabric 1, fabric 2, fabric 3, fabric 4, fabric 5, fabric 5+dw, fabric 5+vb, and fabric 6; the categories of masks are derived from those defined in the literature [101]. (**c**) Basic filtration efficiency of mask material [87]. (**d**) The filtration efficiency with insertion of a layer into a 2-layer cotton mask. T-1–T-16 represents cotton 1 layer, dust mask, #4 coffee filter, cotton 1 layer, shop towel, filtrate 1500, surgical warp, N95 1 layer, medical mask, shop vac, KN95, N95 2 layers, 3M-8511, FTR467 ULPA, dusk mask, and sealed KN95 [87].

**Figure 4 ijerph-20-02346-f004:**
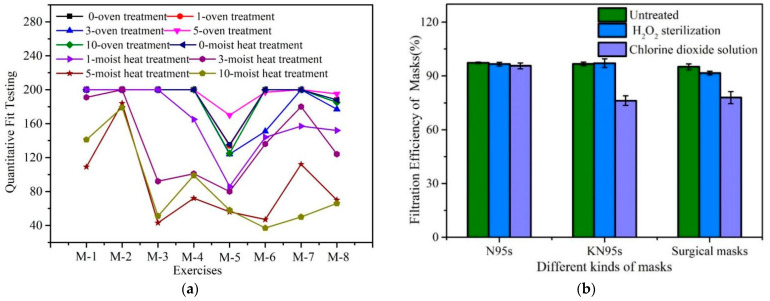
Fitness and decontamination research on filtration efficiency: (**a**) quantitative fit testing results for the new N95 respirator and after oven and steam heat-treatment cycles; M-1-M-8 represent normal breathing, deep breathing, head side to side, head up and down, talking, bending over, normal breathing, and overall fit factor, respectively [103]; (**b**) filter efficiency of mask after hydrogen peroxide and chlorine dioxide decontamination [102].

**Figure 5 ijerph-20-02346-f005:**
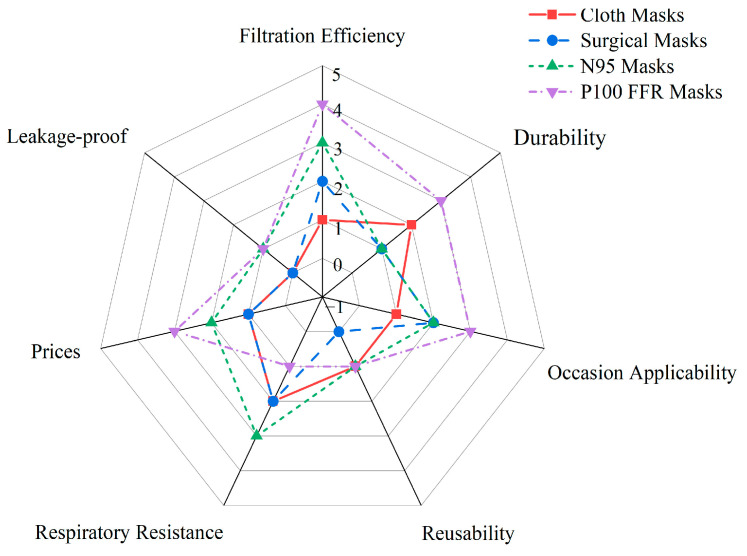
The filtration performance evaluation of masks.

**Figure 6 ijerph-20-02346-f006:**
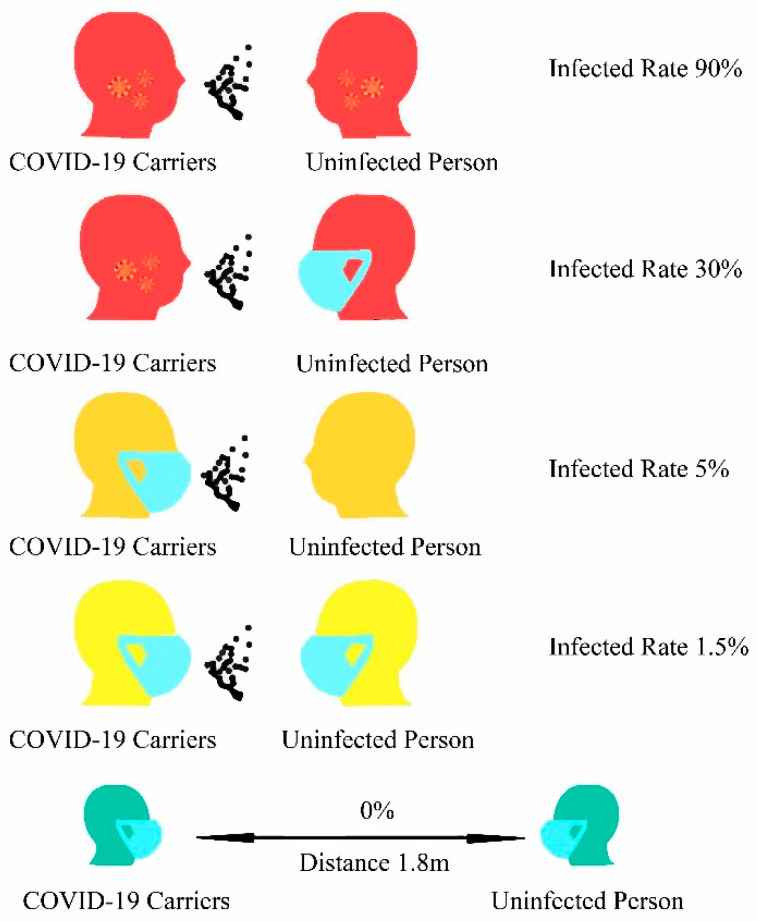
The infection rate when healthy people come into contact with COVID-19 carriers.

**Table 1 ijerph-20-02346-t001:** Aerosol mechanism type, size, and driving force.

Mechanism	Particles Size Range	Main Force
Gravity sedimentation	1–10 μm	The force of gravity
Inertial impaction	>1 μm or larger	Inertia force
Diffusion	<0.2 μm (20 nm–400 nm)	Concentration gradient
Interception	<0.6 μm	Van der Waals force
Electrostatic attraction	Large or small size	Electrostatic attraction

**Table 2 ijerph-20-02346-t002:** A summary of some experimental studies on filtration performance [53,86,87,88,89,90,91,92,93,94,95,96,97,98,99,100,101,102,103].

Filter Materials/Masks	Test Conditions	Particle/Size Range	Reference
Cotton, silk, chiffon, flannel, various synthetics, and their combinations	TSI Nanoscan analyzer TSI OPS analyzer	NaCl aerosol <300 nm and >300 nm	[53]
Several homemade masks and surgical masks	Healthy volunteers Several air-sampling techniques Flow rate of 30 L/min *p* < 0.01 and 0.001	Bacillus atrophaeus 0.95–1.25 μm Bacteriophage MS2 23 nm	[86]
16 single or multilayer fabric or mask test materials	Nanoscan 3910 scanning mobility particle sizer/TSI Portacount Flow rate: 0.0465 LPM/cm^2^ (Light activity) Background particulate count: <10 particles/cm^3^ Cleanroom class: 1000	A polydisperse silicon dioxide nanoaerosol 60–125 nm	[87]
11 types of N95 respirators and surgical masks	ARTI Hand Held Particle Counter/AGI-30 samplers collect endospore bioaerosol samples Flow rate: 594 L/min	A Collision nebulizer aerosolized B/anthracis Sterne strain endospores and polystyrene latex (PSL) particles. PSL particles 0.1 μm, 0.43 μm, 0.6 μm, 1.3 μm, 3.2 μm, or 8.0 μm.	[88]
A. Medical mask, B. medical surgical, And C. newly laundered cloth SM	TSI Automated Filter Tester 8130 (PRC National Standard GB/19083-2010 Temperature: 70 ± 3 °C–−30 ± 3 °C Flow rate: 85 L/min *p* < 0.05	NaCl aerosol 0.075 μm	[89]
Mask 1-11 Outer and Mask1-11 Lining	TSI Model 8130A ASTM F3502 Standard Test Method Flow rate: 85.0 L/min Temperature of 20 ± 2 °C RH: 65 ± 2% *p* < 0.5	NaCl aerosol 0.26 µm Poly alpha olefin (PAO) oil aerosol mass median particle 0.33 µm	[90]
HDX H950, AX-KF95, NANO KN95, ARUN KN95, MERV 13-AIRx, MERV 13-H, HEPA, WypAll X80, and Halyard H600	Portable Aerosol Spectrometer (PAS) GRIMM 11-D system (Particle Counter) TSI Portacount 8030 unit. Flow rate: 2.5 cm /s and 7.1 cm/s 17.4 cm^2^ material area	Polystyrene latex particles 0.25–35 μm	[91]
N95, Surgical 1, Surgical 2, Fabric 1, Fabric 12, Fabric 3, Fabric 4, Fabric 5, Fabric 5 + dried baby wipe, Fabric 5 + vacuum cleaner bag, Fabric 6, Disposable 1	ASTM F2101-14 standard test method	Bacteriophage MS2 6.0 μm and 2.6 μm	[92]
Homemade masks and medical masks	ASTM standard test method Scanning mobility particle-size spectrometer	NaCl aerosols 6–220 nm	[93]
N95-1, N95-2, surgical-style, and other masks	TSI PortaCount TSI Particle Generator Model 8026 65 m^3^ rectangular room	Solution of sodium chloride (NaCl) 5.6 to 560 nm 0 0.3~10 μm	[94]
N95 filtering facepiece respirator and one surgical mask	Twenty-five subjects were selected for NIOSH fit test Utilizing a novel breathing recording and simulation system Flow rate: 10–20 L/min	NaCl aerosols 0.03–1 µm	[95]
Surgical wraps	ASTM environmental laboratory Bacterial, latex particle, and delta P filtration efficiency testing	Latex particle and delta P	[96]
9 types of surgical masks	NIOSH test evaluated for facial fit with volunteers, using both qualitative and quantitative fit tests with direct reading, light scattering photometer Automated Filter Tester Flow rates: 6 L/min and 84 L/min	Monodisperse latex sphere aerosols: 0.895, 2.0, and 3.1 mm Sodium chloride particles: 0.075 mm	[97]
N95 respirators and surgical masks	ASTM test for volunteers Temperature: 25 °C Relative humidity: 70%	Potassium chloride (KCl) solution With concentrations of 0.02, 0.01, 0.005, 0.0025, and 0.0012	[98]
29 different fitted face mask alternatives, such as 3M 1860 N95 respirators, surgical mask with ties, and others	Occupational Safety and Health Administration’s Quantitative Fit Testing Protocol Particle Generator 8026 (TSI) Temperature: 23 °C–29.5 °C Relative humidity: 10%–50%	NaCl aerosols 0.02–3.00 μm	[99]
Respirator, N95, small face type, cup faced, surgical, facemask, and others	A scanning mobility particle sizer (model # 3936, TSI Inc.) along with a condensation particle counter (model # 3788, TSI Inc., USA) Flow rate: 1–85 L/min	Charge-neutralized, dried, and polydispersed sodium chloride (NaCl) 18.8–710 nm	[100]
Sweatshirts, T-shirts, towels, scarves, and cloth masks N95 respirator	TSI 8130 and TSI 3160 Flow rates: 5.5 and 16.5 cm/s	NaCl aerosol particles 20–1000 nm	[101]
N95s, KN95s, and surgical facemasks	Quality Improvement Reporting Excellence (SQUIRE) reporting guideline	NaCl solution 10–1000 nm	[102]
3M 8210 respirator, Halyard 48207 surgical mask, 3M 1820, 2 face masks, respirator materials, Halyard H600 sterilization wrap, and Cummins EX101	Fractional efficiency measurement method Flow rate: 85 L/min	NaCl nano- particles 30–400 nm	[103]

**Table 3 ijerph-20-02346-t003:** The results for the filtration efficiency of five kinds of common fabrics with varied face velocities [101].

Face Velocities (cm/s)	Cloth Mask (FE%)			
1	Media N95	Respro Bandit	Breathe Health Cloth	Breathe Health Fleece
5.5	99.88%	10%	15%	25%
16.5	96.60%	11%	18%	25%
2	Sweatshirt			
	Media N95	Normal kamali	Hanes	Faded Glory
5.5	99.88%	32%	60%	16%
16.5	96.60%	25%	42%	17%
3	T-shirt			
	Media N95	Dickies	Hanes	Faded Glory
5.5	99.88%	12%	13%	88.50%
16.5	96.60%	9%	10%	10%
4	Towel			
	Media N95	Pem America	Pinzon	Aquis
5.5	99.88%	37%	40%	33%
16.5	96.60%	36%	38%	30%
5	Scarf			
	Media N95	Today’s gentleman	Walmart	Seed Supply
5.5	99.88%	10%	28%	10%
16.5	96.60%	12%	22%	8.50%

## Data Availability

The datasets used during the current study are available from the corresponding author upon reasonable request.

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
