# Peer review of "A Review of Filtration Performance of Protective Masks"

_ijerph, 2023, doi:10.3390/ijerph20032346_

Round 1

Reviewer 1 Report (New Reviewer)

I thought that this manuscript could make a great publication after I looked at Figure 2.Unfortunately,it did not go further.This review is confusing and comes after very confusing publications: 1) T Jefferson et al .Physical interventionsto interrupt...Cochrane Library 2020; 2) M Loeb et al Medical masks versus N95 respirators...Ann Intern Med 2022.

This manuscript shall be structured and rely on a sound litterature review such as:. 1) CC Wang et al Airborne transmission of respiratory viruses.Science 2021. 2) Y Wang et al. Modeling the load of SARS...PLOS One 2020. 3) PZ Chen et al. Heterogeneity in transmissibility and shedding.....eLIFE 2021 .4) CR MacIntyre et al .A systematic review of the efficacy..Int J Nursing Stud. 2020. These publications allow to better understand why the population needs masks and how they work.

     Good descriptions of masks are given in the following publications: 1) CM Sterr et al Medical face masks offer self protection...PLOSOne 2021. 2) M Zhao et al Household materials selection...NanoLetters  2020. 3) X Guan et al Prolonged use of surgical...Materials 2022.

      This review is very meritorius but boring for the readership.The authors have demonstrated in Figure 2 their capacity to describe  concepts ,they must pursue in this direction.They also can benefit of Radar Charts' representation to figure out the tables.

     This review shall be made narrative to atract the readership and answers the following issues; 1) What are the needs in the hospital,on the street,during an operation,etc...2) What is available to-day on the market ? Conclusion: am I safe in this world? Hope yes: why?

       This paper needs a major revision.

Author Response

    We appreciate the helpful and insightful comments from the reviewers. All the team members speak highly of your responsible attitude in reviewing our paper. We believe that we have addressed all the comments through which we have improved the quality of our manuscript.

    Note that if the modifications made according to your comments only, yellow will be used in highlighting this revision. If the modifications made according to more than one reviewer, green will be used. Thank you very much. 

Responses against each point can be found in the word  files.

Reviewer 2 Report (New Reviewer)

Dear Authors.

I congratulate you on an interesting idea and detailed analysis of the issue.

The main note to the paper is the inconsistency between the review of the filtration mechanism, test methods, etc. (up to line 438) and the filtration efficiency of the masks (from line 439).

I believe that the exact purpose of the study should be indicated and then the text of the paper should be selected. First, the types of masks and their purpose should be presented, and then other information.

Moreover:

1.      Issues related to the development of COVID are not related to the scope of the paper, so lines 36-61 should be deleted.

2.      Authors often present the use of masks as protection against COVID. In fact, this is just one of the many factors that masks protect against, so a wider range of protection should be provided.

3.      In lines 86-87 the Authors refer to Appendix A. I did not find the Appendix in the materials provided.

4.      Chap. 2.1 should be described according to the systematics, e.g. due to the purpose of the mask determined by the type of hazard.

5.      Line 130 "In this case, ...". What case do the authors describe?

6.      Line 96 "there is a lack of systematic research on the filtration mechanism, ...". On what basis is such a claim, since there are many publications on this subject, and the methodology is defined by standards?

7.      Line 300-303 - Why was the verbal communication problem omitted?

8.      Line 306 "At present, the researches focus ..." - Which research?

9.      Chap. 3.1 - Why are test methods presented referring to selected types of masks and not factors? What was the criterion for selecting the presented content?

10.   On what basis was the chapter 4.5? The text does not result from the content of the study or comparative results.

11.   Lines 267-637 - I agree with this conclusion, but it does not follow from the content of the paper. The results of the comparative analysis in this respect have not been presented. Only different mechanisms are shown, not their interactions. The request should be deleted or described differently.

Author Response

Your painstaking work in reviewing our paper is highly valued. Your suggestions have made a great improvement in our paper.

Note that if the modifications made according to your comments only, purple will be used in highlighting this revision. If the modifications made according to more than one reviewer, green will be used. Thank you very much.

Responses against each point can be found in word files.

Round 2

Reviewer 1 Report (New Reviewer)

Congratulations! What a difference with the previous version! 

Reviewer 2 Report (New Reviewer)

Dear Authors

Thank you for the changes made.

I accept all changes.

Good luck

This manuscript is a resubmission of an earlier submission. The following is a list of the peer review reports and author responses from that submission.

Round 1

Reviewer 1 Report

Dear Authors, 

Thank you very much for your contribution and hard work on the topic of protective masks.

In my opinion, the paper can be published ONLY if it undergoes significant changes. Unfortunately, it seems that the authors tried to pack a more extensive set of information, perhaps coming from a project or a Ph.D. thesis. Therefore, to improve the readership of this manuscript, I suggest the following changes:

1) Reduce the amount of text in your manuscript and limit the scope while improving in-depth analysis.

2) The current manuscript contains at least four different types of studies combined, making it very difficult to follow. Therefore, I see four options here:

a. You can shape this paper as an introduction to the study on masks and publish only the content presented on pages 1- 11.

b. Next, you can publish a comparative study of masks, where you focus on some selected parameters of masks.

c. Physiological study/human comfort should be published as a separate study

d. The final paper can discuss everything else which is not under a), b), and c)

3) Since you try to push a large set of information, sometimes you tend to be superficial or use a shortcut rather than showing an in-depth analysis of the subject. For example, it is obvious when introducing new terms, page 3, lines 82-92. However, this is true for an entire paper when you present several terms, one after another, without defining them, e.g., nanofiber, nanoparticles, the Brownian motion, etc.

4) All the images must be bigger as, currently, it is difficult to see/read their content.

5) The title and the abstract need to reflect the content of your work. 

6) In Figure 3, the selection of bacteria types, reasoning, and background is unclear, especially since you started from the corona virus.

Reviewer 2 Report

The manuscript discusses the Investigation and Efficiency Research on Different Scenarios of Mask Protection Materials against COVID-19. The topic is interesting and has merit. The paper reads well. I have some minor comments:

-          - Please provide critical discussion for the data.

-        -   Provide more conclusive findings in the abstract.

-         -  Enhance your literature by citing relevant studies.

-          Examples: - https://doi.org/10.3390/su14020737

-          https://doi.org/10.1016/j.cscee.2021.100151

-         - In all Tables and Figures- Please provide proper citations and comparison to other previous works where applicable.
